# Direct space–time manipulation mechanism for spatio-temporal coupling of ultrafast light field

Qinggang Lin[1,3], Fu Feng[1,2,3], Yi Cai[1], Xiaowei Lu[1], Xuanke Zeng[1], Congying Wang[1], Shixiang Xu ®[1] ✉, Jingzhen Li[1] & Xiaocong Yuan ®[1,2] ✉

Traditionally, manipulation of spatiotemporal coupling (STC) of the ultrafast light fields can be actualized in the space-spectrum domain with some 4-f pulse shapers, which suffers usually from some limitations, such as spectral/pixel resolution and information crosstalk associated with the 4-f pulse shapers. This work introduces a novel mechanism for direct space-time manipulation of ultrafast light fields to overcome the limitations. This mechanism combines a space-dependent time delay with some spatial geometrical transformations, which has been experimentally proved by generating a high-quality STC light field, called light spring (LS). The LS, owing a broad topological charge bandwidth of 11.5 and a tunable central topological charge from 2 to −11, can propagate with a stable spatiotemporal intensity structure from near to far fields. This achievement implies the mechanism provides an efficient way to generate complex STC light fields, such as LS with potential applications in information encryption, optical communication, and laser-plasma acceleration.

Spatiotemporal coupling (STC) light fields[1], with inseparable space and time distributions, have garnered great interest in fundamental studies of space–time light manipulation over the past decade[2–8]. They include ultrashort pulse velocity control[9], nonlinear optics[10], and others. Furthermore, such light fields have been demonstrated with many potential applications, e.g. the generations of free-space accelerated and decelerated optical wave packets[11], light bullets[12], and non-diffracting optical vortices[13,14].

However, as ultrafast light fields evolve so rapidly that light modulators are unable to cope with them, the STC manipulations of light fields are usually performed in the space-spectrum domain to avoid direct space–time manipulation by means of the Fourier correlated nature of time and spectrum. In this case, a 4-*f* pulse shaper comprised of a pair of gratings and a 4-*f* imaging system[15,16] has been widely used to realize STC of light fields. At the intermediate confocal plane of the 4-*f* imaging system[17–19], the incident spectral components are separated spatially to be manipulated by an optical modulator, e.g., a spatial light modulator[11,13,20–23] or phase plate[24,25]. This kind of space-spectrum manipulation has facilitated to generate a series of interesting STCs of light fields, including spatial frequency-time accelerated or decelerated wave packets[11], beams holding time-varying orbital angular momentum[26], and vortex wave packets with spiral phase[14,16–19,23]. Although the 4-*f* pulse shaper has achieved great success, it encountered a couple of intrinsic problems. Firstly, it suffers from poor spectral resolution because of either the limited apertures of the phase plates or the poor pixel resolution of the spatial light modulators; this becomes even more critical when large spectrum-dependent delays are required to generate light fields with some special space–time trajectories, such as the single-coil light spring (LS)[27,28]. Secondly, manipulation at the confocal plane will cause spectrum-dependent diffraction and the resultant information crosstalk, significantly degrading the quality of the generated light field[29]. In

[1]Key Laboratory of Optoelectronic Devices and Systems of Ministry of Education and Guangdong Province, College of Physics and Optoelectronic Engineering, Shenzhen University, 518060 Shenzhen, China. [2]Research Center for Humanoid Sensing, Zhejiang Laboratory, 311100 Hangzhou, China. [3]These authors contributed equally: Qinggang Lin, Fu Feng. ✉e-mail: shxxu@szu.edu.cn; xcyuan@szu.edu.cn

addition, the gratings could also be problematic due to limited diffractive efficiency and fabrication difficulties for certain spectral regions.

This paper proposes a novel mechanism for direct space–time manipulation. It begins by introducing a tilted pulse-front of an ultrafast light field, thereby producing a space-dependent time delay, then performs a geometrical transformation to spatially shape the pulse-front tilted laser. The method eliminates the need to separate spatially the laser spectrum, thus avoiding the problems associated with optical grating. Moreover, the pulse-front tilted pulse, created through angular dispersion, exhibits continuous spatiotemporal variation, resulting in a high-quality beam for the targeted STC of light field. To validate the proposed mechanism, an exemplary experiment to generate a single-coil LS is designed and conducted. Measurements show that the single-coil LS possesses a tunable topological charge (TC) with a broad TC bandwidth, leveraging great potential applications in laser-plasma dynamics[30–32], information encoding[33] and laser-driven electron acceleration applications[34,35]. Notably, the unique angular delay of the single-coil LS can serve as the light source for annular on-chip dielectric laser-based accelerators. To the best of our knowledge, a single-coil LS with a wide TC bandwidth has never been realized experimentally. The proposed mechanism and method will pave the way toward the generation of complex STC fields and shows great promise for a range of practical applications.

## Results

### The concept and theoretical realization

The basic concept of direct space–time manipulation involves first to create a space–time correlation within a light field, followed by executing the corresponding spatial transformations to achieve the targeted STC light field. For simplicity, in this work, the first step only links time to one-dimensional space, and subsequent spatial transformation converts this correlation into a correlation between time and two-dimensional space. Assuming that an ultrafast light field has a linearly space-dependent time delay, the field can be described as

$$E_1(x,y,t) = A(x,y) \cdot \exp\left[-\alpha(t - \eta y)^2\right] \cdot \exp[i\omega_0(t - \eta y)]. \quad (1)$$

Here, $A(x,y)$ denotes the spatial amplitude distribution of the incident light field, and $\alpha = C/\tau_p^2$ is determined by the characteristic constant $C$ and the pulse width $\tau_p$. $\eta y$ represents the time delay along the $y$-axis, which is proportional to the factor $\eta$ and the coordinate $y$. After the spatial transformation, the light field can be expressed as

$$E_2(u,v,t) = E_1[x(u,v),y(u,v),t], \quad (2)$$

where the coordinates $(x, y)$ and $(u, v)$ are related through a geometrical transformation. In our design, the space-dependent time delay is realized by introducing a tilted pulse-front through an angular dispersion optical element, such as a prism. Meanwhile, the geometrical transformation is implemented using a log-polar-Cartesian transformation[36] to generate the single-coil LS.

If the seed light field is a fundamental-mode Gaussian (TEM$_{00}$) femtosecond pulse, after passing through the angular dispersion element, the ultrafast light field with the tilted pulse-front can be expressed as

$$E_{PLT}(x,y,t) = A_0 \cdot \exp\left(-\frac{x^2}{r_x^2} - \frac{y^2}{r_y^2}\right) \cdot \exp[-\alpha(t - \eta y)^2] \cdot \exp[i\omega_0(t - \eta y)], \quad (3)$$

where $A_0$ is the amplitude, and $r_x$ and $r_y$ correspond to the beam sizes along the $x$ and $y$ directions, respectively.

The log-polar-Cartesian transformation system includes a geometric transformation element (GTE) and a phase compensation element (PCE), which are positioned at the front and rear focal planes of an optical lens, respectively. For an incident light field with a tilted pulse-front, under the stationary phase approximation[37], the output light field from the log-polar-Cartesian transformation system can be expressed as

$$E_{LS-t}(u,v,t) = \frac{2\pi aA}{b} \exp\left(\frac{r_x^2}{4a^2}\right) \cdot f(u,v) \cdot g(u,v,t)$$
$$\cdot \exp\left(i\omega_0 t + il_0 \cdot \arctan\left(\frac{v}{u}\right)\right). \quad (4)$$

in the space–time domain, and

$$\tilde{E}_{LS-f}(u,v,\omega) = \frac{2\pi aA}{b} \sqrt{\frac{\pi}{\alpha}} \exp\left(\frac{r_x^2}{4a^2}\right) \cdot f(u,v) \cdot q(u,v,\omega), \quad (5)$$

in space-spectrum domain (see details in Part I of the Supplementary Material). Here, $l_0$ is a constant topological charge (TC) in time domain, which is related to $a$, $v$-directional relative displacement between GTE and PCE apart from the focal length of the optical lens, and

$$f(u,v) = \exp\left[-\frac{a^2}{r_x^2}\left(\frac{r_x^2}{2a^2} + \ln\frac{\sqrt{u^2 + v^2}}{b}\right)^2\right] \cdot \exp\left[-\frac{a^2}{r_y^2}\arctan^2\left(\frac{v}{u}\right)\right], \quad (6a)$$

$$g(u,v,t) = \exp\left[-\alpha\left(t - \eta a \cdot \arctan\left(\frac{v}{u}\right)\right)^2\right] \quad (6b)$$

and

$$q(u,v,\omega) = \exp\left(\frac{-(\omega - \omega_0)^2}{4\alpha}\right) \cdot \exp\left(-i(\omega - \omega_0)\eta a \cdot \arctan\left(\frac{v}{u}\right)\right). \quad (7)$$

Equation (4) displays a spiral phase with a constant TC in the space–time domain of the generated light field. Equation (6a) depicts the annular-shaped and azimuth angle-dependent amplitudes, while Eq. (6b) presents an azimuth angle-dependent linearly time delay with a rotation period of $2\pi\eta a$. The exponential expansion term in Eq. (6b) includes a time-space product, indicating that the single-coil LS is a STC light field. Accordingly, Eq. (5) demonstrates that both the amplitude and phase of the light field have spiral-like structures. In the space-spectrum domain, Eq. (7) reveals another interesting characteristic of this light field: linearly spectrum-dependent TC values, with a TC bandwidth of $2\pi\eta a \cdot \Delta v$.

### Experimental realization with an exemplary result

An experimental setup has been designed to realize the proposed direct space–time manipulation, as detailed in the Method section. At the moment, the space-dependent time delay is introduced by tilting the pulse-front of an ultrafast light field using a Brewster prism, which would lead to the coupling between one-dimensional space and time. Subsequently, a log-polar-Cartesian transformation is designed to transform the coupling between one-dimensional space and time into that between two-dimensional space and time. In the setup, the light source is a fiber laser which outputs 30.58 MHz/172 fs/1030 nm linearly polarized pulses. The Brewster prism is made of H-ZF13 glass, and the designed phase modulations, including the GTE and the PCE for log-polar-Cartesian transformation, have been experimentally verified separately, as shown in Fig. S1 in Part I of the Supplementary Material. As indicated in Fig. S1, the spatiotemporal manipulation is continuous, which, consequentially, is one of key factors for the generation of high-quality single-coil LS.

Figure 1a1–a11 present the spatial intensity distributions of the output light field by pump-probe measurement (refer to Fig S4 in Part

II of supplementary material for details) with a time interval of 266.7 fs to sample the light field. Here, the time interval can be changed by adjusting the step size of the time delay line. Figure 1b1–b11 is the corresponding calculated results. It is observable that the experiment results exhibit the same time-varying trajectories as the calculated, specifically, both the beam spots rotate counterclockwise, which is consistent with the temporal variation properties given by Eq. (4). Figure 1c shows the measured spatiotemporal intensity evolution (red, constructed from Fig. 1a1–a11) and the simulation results (cyan). It can be seen that the measured results correspond well with calculations, which confirms the successful generation of the single-coil LS. However, it is worth pointing out that the red image is discrete, which can be attributed to the scanning time interval of the pump-probe measurement (266.7 fs) is chosen to be larger than the pulse width (172 fs) to avoid overlapping between neighbored frames. As mentioned

previously, one can choose a shorter time interval by adjusting the delay time to sample spatiotemporal structure of the light field for finer details in time domain. However, a short time interval can make the pump-probe measurement time-consuming, moreover it will not improve the sampling quality. The good matching between the red and cyan results in Fig. 1c indicate that a time interval of 266.7 fs is short enough to effectively sample the single-coil LS with a rotation period of 2.15 ps. Figure 1a12 and b12 shows the measured and calculated spatial intensity distribution of the generated light field after time integration, respectively. Both figures show a circular profile with a gap located at its bottom and its intensity firstly increases and then decreases with the azimuth angle, demonstrating good consistency between them.

To further quantitatively explore the properties of this light field, the evolution of the rotation angle over time is plotted in Fig. 2a, with the red squares representing the data extracted from experiments and the red line indicating a linear fitting of the red squares. The blue squares and line are corresponding calculated results. From Fig. 2a, the rotation speed can be estimated to be 2.92 rad/ps, and the rotation period Δ$t$ is thus determined as 2.15 ps, which corresponds to the total time needed for a 360° rotation of the light spot in azimuthal direction. Moreover, it can be seen that the rotation period is longer than the seed pulse duration $τ_p$ which is measured to be 172 fs due to dispersion introduced during modulation, as shown in Fig. 2b. Additionally, from the autocorrelation profile, it can be observed that the seed pulse contains a picosecond background, which, together with optical diffraction, contributes to the background noise in Fig. 1a1–a12. The blue squares and line in Fig. 2a represent the corresponding calculation results, which agrees well with the measurements. Based on the above results, it can be concluded that the measurements and the theoretical predictions have excellent agreement, which indicates experimentally a successful generation of a single-coil LS experimentally. According to Eq. (6b), the spatiotemporal information of a single-coil LS cannot be separated, i.e., the LS is space–time coupled[38,39]. Experimentally, time-varying spatial intensity distribution of the single-coil LS, as shown in Fig. 1c, further confirms the STC characteristics of the generated LS.

In this section, measurements are taken and displayed in Fig. 3a1–a12 to assess the spectral TC distribution of the generated LS with the experiment setup shown in Fig. S4. For this measurement, a narrow slit is inserted at the intermediate confocal plane of the 4-$f$ system, by shifting the slit position, twelve narrowband spectra are selected, with the central wavelengths ranging from 1023 nm to 1039 nm with an interval of ~1.46 nm and a bandwidth of ~1.20 nm, as shown in Fig. 3(c). From Fig. 3a1–a12, the spectral TC values are extracted, which varies from −6 to −17 over the wavelength from 1023 nm to 1039 nm, as indicated by the red dots in Fig. 3d. The central

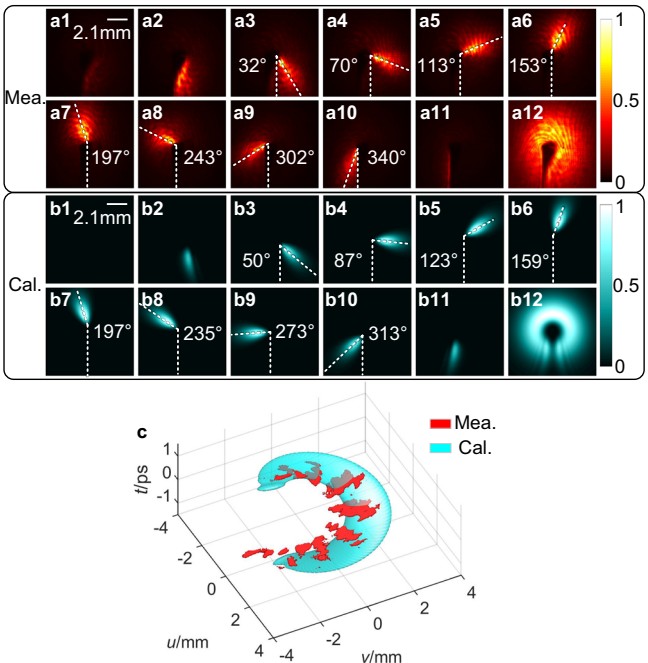

**Fig. 1 | The spatial intensity distributions of the generated light field. a1–a11.** The measured spatial intensity distribution at different time. **b1–b11** The corresponding calculated results. **a12, b12** The measured and calculated spatial intensity distribution after time domain integrations, respectively. **c** The measured (red) and calculated (cyan) spatiotemporal structure of the generated light field.

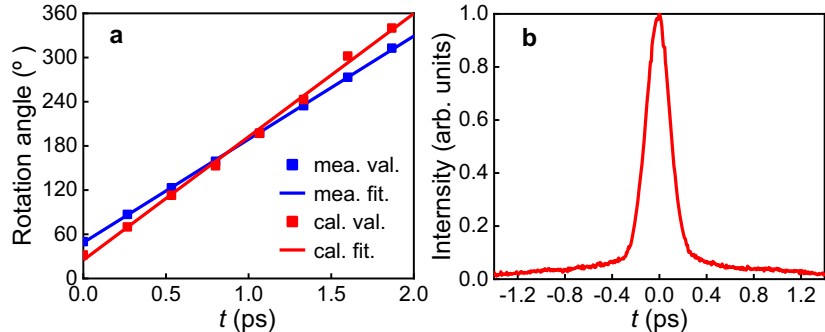

**Fig. 2 | The rotation angle as function of time and the measured pulse duration. a** The measured and calculated rotation angle of the output pulse intensity as a function of time by red dots for the measured and blue dots for calculated results

with the red and blue lines for the corresponding linear fitting, respectively. **b** The temporal profile of the femtosecond pulse from the 1030 nm fiber laser measured by an autocorrelator.

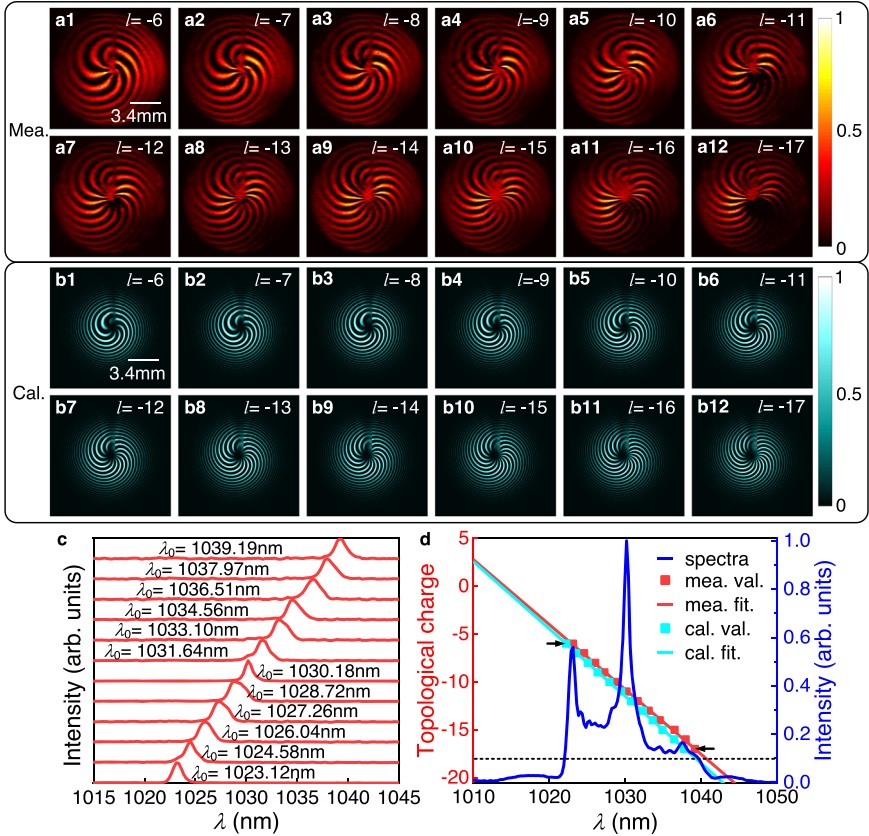

**Fig. 3 | Interference patterns between the output from the log-polar-Cartesian transformation system and the femtosecond probes at twelve selected narrowband spectra. a1–a12** The measured and (**b1–b12**) simulated interference patterns at different narrowband spectra. **c** Measured narrowband spectra selected by the slit at different position. **d** Spectral TCs (red: measured, crayon: calculated) vs. wavelengths with the seed pulse spectrum (blue): the square marks for the results extracted from the interference patterns and the lines for corresponding linear fittings, respectively.

TC, or the TC at the central wavelength, is estimated at −11. Figure 3d also presents the spectrum of the seed laser by the blue curve and the theoretical fitting of the red dots by the red line. One can see that the spectral TC values have a good linear dependence with wavelengths within the effective spectral region of the seed laser. This important property is consistent with the prediction of the spectral TC variation by Eq. (5). The spectral TC changes by a speed of $\Delta l/\Delta\omega$=0.381 THz$^{-1}$, and the TC bandwidth is as broad as 11.5. The corresponding simulations are presented in Fig. 3b1–b12, and the corresponding TC values are extracted as shown by the cyan dots in Fig. 3d where the cyan line is used for a linear fitting of the cyan dots. Here, all the simulations based on the seed with the measured spectrum shown by the blue line. It can be concluded that the spectral TC varies linearly by a speed $\Delta l/\Delta\omega$ = 0.390 THz$^{-1}$, and the TC bandwidth is 11.3. Accordingly, the spectral TC variation obtained from the experiment is consistent well with theoretical expectations. Based on the linear dependence of the spectral TC on wavelength, it can be further demonstrated that the output field constitutes a single-coil LS.

In this section, it is demonstrated that the central topological charge value is tunable. Figures 4a–n are the measured interferograms of 1030 nm LS with PCE at different positions, it is shown that the central TC value of the generated LS can be tuned by changing the position of the PCE plate. Figure 4o plots corresponding central TC values as a function of PCE position (hollow red dots), while blue curve represents a linear fitting of these values. Obviously, the central TC is linearly tunable from 2 to −11 with the y-directional position of the PCE. More preciously, the TC value changes by 1 with 44 μm displacement of the PCE plate, which is consistent with the theoretical

value (45.5 μm) obtained by Eq. (S13) from Part I of the Supplementary Material.

In this section, it is demonstrated when tuning the central TC by changing the position of the PCE plate along the y-direction, the LS keeps its spectral TC bandwidth nearly invariant. Figure 5 shows the measured TC values of the single-coil LS over wavelength for different central TCs (2, 0, −3, and −11). The square solid points and the solid lines represent the experimental results and the fitting curves, respectively. The red, blue, magenta, and cyan colors represent the LSs with central TCs of 2, 0, −3, and −11 with the TC values varying at a speed of ($\Delta l/\Delta\omega$) of 0.387 THz$^{-1}$, 0.389 THz$^{-1}$, 0.375 THz$^{-1}$, and 0.381 THz$^{-1}$ respectively. As a result, the TC bandwidths can be estimated to be 11.7, 11.8, 11.3, and 11.5, correspondingly, which shows that tuning of the central TC makes nearly no change to the TC bandwidth.

At the end, the propagation properties of the LS have also been investigated. Figure 6 shows the measured results of spatiotemporal intensity distributions of generated LS at both near-field and far-field. Figure 6a1–a11 exhibits the spatiotemporal intensities over time at focal plane, while Fig. 6b1–b11 represent the corresponding distributions at the plane which is 2 cm behind the focal plane, respectively. The measured rotation periods of the field at these two planes are estimated to be 2.14 ps and 2.34 ps, respectively, which are consistent with the rotation period (2.15 ps) at the near field. It conforms to the theoretical expectations and the propagation characteristics for the LS in reference 3. Figure 6a12 and b12 shows the measured and calculated spatiotemporal intensity after integration over time. When comparing these two cases, it can be observed that an intensity gap appears at the bottom of the ring on the plane 2 cm behind the focal plane, which can

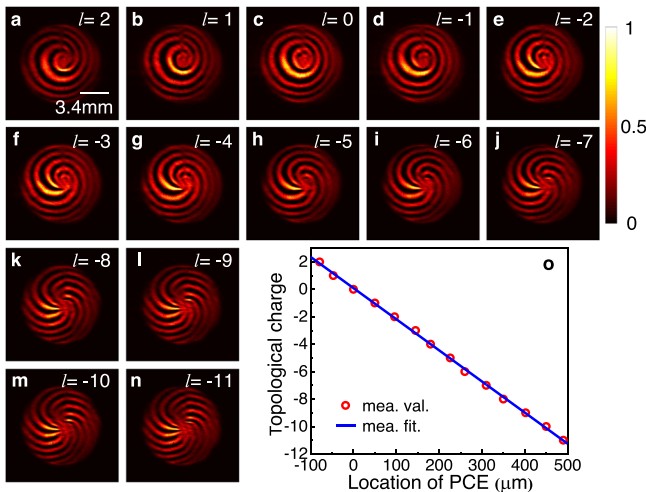

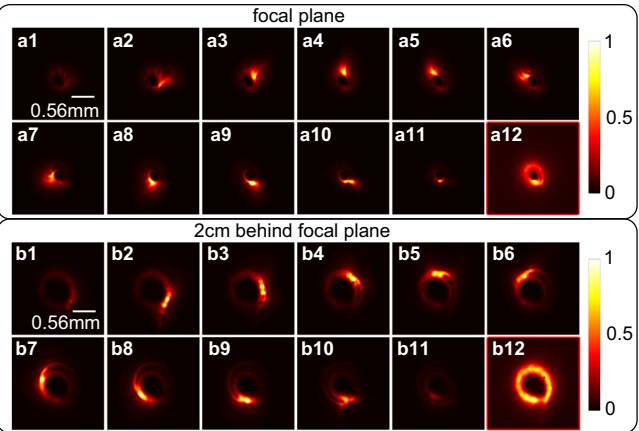

Fig. 6 | The Spatiotemporal intensity distributions of the generated light field after different propagation distance. a1–a11, b1–b11 Recorded spatial intensities of the generated single-coil LS as a function of time at the focal plane and 2 cm behind the focal plane, respectively. a12, b12 The corresponding spatial intensities distribution after integration over time.

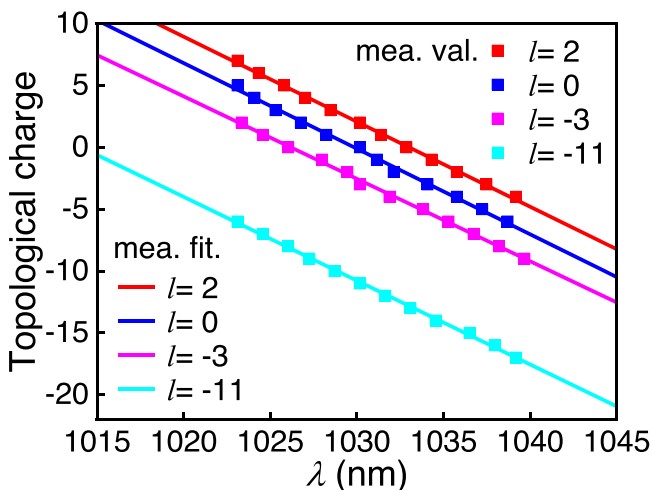

Fig. 4 | Central topological charge of the LS with respect to the PCE position along y-direction. a–n Measured interference patterns with different PCE positions. o Measured central TCs of the interference patterns as a function of PCE position by the red hollow points are extracted from the interference patterns (a–n), and the blue line corresponds to the fit curve.

Fig. 5 | Spectral TC values of the single-coil LS vs. wavelength for different central TCs (2, 0, −3, and −11). The square solid points and solid lines represent the experimental results and the fitting curves, respectively, where the red, blue, magenta, and cyan colors indicate the TC distributions of the single-coil LS for the central TCs of 2, 0, −3, and −11, respectively.

be attributed to optical diffraction. Nevertheless, the results suggest that the LS can maintain its helical structure while propagating in free space, which is critically important for further manipulation and application.

## Discussion

The experimental results have confirmed the effectiveness of the proposed direct space–time manipulation in realizing STC LS, which is previously unachievable with other methods. Figure 1 showcases its spring-like amplitude, while Fig. 3 demonstrates that the spectral TC values of generated LS vary linearly with wavelength, which is one of the key unique properties of single-coil LS. Moreover, the generated LS has broad TC bandwidth in spectral domain, which, according to Eq. (7), is proportional to the seed pulse bandwidth $\Delta\omega$, angular dispersion coefficient $\eta$ and the beam size $a$. From Figs. 4, 5, it can be seen

that the central TC value of the single-coil LS can be tunned continuously by changing the position of the PCE plate along y-directional while the TC spectral bandwidth is maintained during this manipulation. This behavior agrees well with theoretical prediction (see Eq. (S13) in Part I of the Supplementary Material). As is known, for a STC light field, the propagation stability is critical for its practical application. Interestingly, as shown in Fig. 6, the measurement results have verified that both the intensity structures and rotation periods of the single-coil LS have excellent propagation stability.

In summary, the generated single-coil LS has a spectral TC bandwidth $\Delta l$ up to 11.5 with its central TC tunable from 2 to −11. The rotation period is measured to be 2.15 ps. Such high-quality single-coil LS has never been experimentally realized. The successful realization is ascribed to the proposed direct space–time manipulation. Compared with classical space-spectrum manipulations based on 4-f pulse shapers, instead of separating spatially the laser spectrum, the proposed mechanism continuously manipulates the ultrafast light field in time domain, thus avoiding the detrimental effects of the poor spectral resolution caused by the restrictions on the clear apertures and the pixel resolution of the phase modulators, as well as information crosstalk caused by optical diffraction during phase modulation. The experimental setup is simple and compact, and it is assembled with all-transmission optical elements, which allows it to work with strong stability and ease the alignment.

The superior characteristics of the single-coil LS in terms of its broad TC bandwidth, tunable central TC, and stable propagation capability, illustrate the significant advantages of the method and its great potential for use in complex STC light field generation. This work provides a novel way to produce the single-coil LS and other types of novel STC beams, which may be applicable to optical communications, information encryption, and laser-plasma acceleration.

## Methods
### Experiment setup to generate single-coil light spring

Figure 7 shows the experimental setup to perform the direct space–time manipulation, which enables the generation of the single-coil LS from an ultrafast light field. A fiber laser outputs 30.58 MHz/ 172 fs/1030 nm pulses with linear polarization. The pulse-front of the femtosecond pulses is tilted using a Brewster prism (H-ZF13) which has negligible inserted loss for the linearly polarized incident pulses. After the prism, a polarizer is used to further purify the polarization of the light field. In order to increase the y-dependent time delay, a pair of column lenses CL1 and CL2 with focal lengths of 50 mm and 500 mm

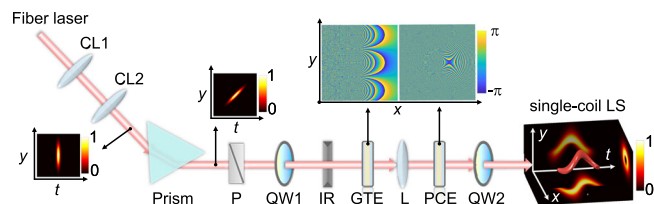

**Fig. 7 | Setup based on direct space–time manipulation to generate single-coil LS from an ultrafast light field.** CL1 and 2: column lenses; Prism: Brewster prism; P: polarizer; QW1, 2: quarter wave plates; IR: iris; GTE: geometric transformation element; L: lens; PCE: phase compensation element. The 3D image of the single-coil LS includes its projection images in the $x$–$y$, $t$–$y$, and $x$–$t$ planes.

are used to expand the beam size of the pulses from 1.8 mm to 18 mm (full width at half maximum, FWHM) along the y-direction. After that, a quarter-wave plate (QW1) converts the pulses from linear to circular polarization. In the log-polar-Cartesian transformation system, the geometrical phase elements, GTE and PCE, made of liquid crystal polymers, are located at the front and rear planes of a lens $L$ with a focal length of 500 mm, respectively. The designed phase modulations of the GTE and the PCE are described by Eq. (S1$a$) and (S1$b$) and confirmed experimentally, as shown in Fig. S1 in Part I of the Supplementary Material. The PCE also acts as a regulator to adjust the parameter $l_0$, which represents the TC value in time domain. Finally, a quarter-wave plate (QW2) is used to convert the polarization from the circular back to the linear, while ensuring it is orthogonal to that of the seed pulses.

The resulting pulse-front tilted pulse exhibits continuous spatiotemporal variation, facilitating the generation of high-quality, linearly polarized STC light fields. Furthermore, geometrical transformations, like the log-polar-Cartesian transformation, can be employed to further modulate the pulse-front tilted laser pulse, enabling the generations of a variety of complex STC light fields.

## Data availability

The data that support the findings of this study are available from https://figshare.com/s/d4b4b0f557f336be91fe.

## Code availability

All code used in this study is available from the corresponding authors upon request.

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

## Acknowledgements

This work is partially supported by Guangdong Major Project of Basic Research (grant No. 2020B0301030009), National Natural Science Foundation of China (Nos. 92050203, 62075138, 12174264, 61827815, 12004261, 62275163); Natural Science Foundation of Guangdong Province (Nos. 2021A1515011909 and 2022A1515011457); Shenzhen Fundamental Research Program (Nos. JCYJ20200109105606426, JCYJ20190808164007485, JCYJ20210324095213037, JCYJ20190808121817100, JCYJ2019 0808143419622 and JCYJ20190808115601653), Shenzhen key technology projects (Nos. JSGG20191231144201722 and JSGG20211108092800001).

## Author contributions

Q. Lin, F. Feng, and S. Xu conceived the concept and design of the experiments. Q. Lin, F. Feng, and Y. Cai carried out the numerical analysis. Q. Lin, Y. Cai, X. Lu, and X. Zeng discussed the results. C. Wang prepared the figures. S. Xu and X. Yuan supervised the entire work, and took part in the preparation of the manuscript together with J. Li. All authors reviewed the manuscript.

## Competing interests

The authors declare no competing interests.
