## [Peer Review File · Nature Communications]

REVIEWER COMMENTS

Reviewer #1 (Remarks to the Author):

I have read and checked as far as I can the manuscript. It seems to me highly interesting as a novel method of sculpturing light, exemplified in the realization of a single-coil spring of light. The differences with other methods are stressed in the introduction. To my knowledge taking advantage of the pulse front tilt has not been proposed previously. The methodology is sound and the work meets the standards in this field.

This work may have a higher impact in the structured light community and in their broad applications if the authors demonstrate, or make it evident, that the same or similar technique can be applied to sculpture spatiotemporal coupled fields other than the single-coil spring of light to which the whole manuscript is focused, and is not limited to light springs. The authors mention "to construct various complex STC light fields" without further development. For example, why only single-coil and not more coils? What about toroidal pulses or STOVs? Which are the limitations of the method? Depending on this, the claim of the title should be lowered.

Other points:

It is a bit heavy to repeat everywhere STC.

The English language needs to be slightly improved. E.g. "one decade due to their many potential [2, 3], ultrashort pulse velocity control..." does not make sense to me.

The references to springs of light are not properly cited in the text, are disordered, and at least one is missing. The first proposal is not in [10], but in PRL 111, 083602 (2013) in the context of high-harmonic generation, which is not cited. Then, quite speculative theoretical considerations are proposed in [10]. Ref. [3] describes the four dimensional structure of springs of light including their propagation properties and, in my opinion, cannot be cited only at the very beginning of the text mixed with other references on other topics. Indeed, the authors observe the propagation properties of light springs as predicted in [3]. Please remove "initially proposed in 2015" in the introduction and in the discussion section (which, in passing, is quite repetitive).

All contour plots in the figures lack titles and units. I suppose the titles are x and y, but it must be written. I cannot get an idea of the size of the springs of light without units.

Other point is that the rotation period is 2150 fs, each frame in Fig. 1 takes 266.7 fs and there are 10 frames, which yields 2667 fs, not 2150 fs. If I'm not wrong, please clarify.

In short, in addition to the above points, the main point is that the authors should expand on the applicability of their method to synthesise other STC fields.

Reviewer #2 (Remarks to the Author):

This paper was written toward a very hot topic, the spatiotemporal light field manipulation. The authors present a beautiful spiral light pulse generation and control with very high tunable topological charges, which looks very nice. While, the authors claim they can do direct spatiotemporal manipulation of space-time coupling of ultrashort pulses, I am afraid the authors didn't really realize this.

Firstly, the authors didn't measure the space-time coupling of the light fields. To claim the space-time coupled light fields, we need to quantitatively measure the degree of space-time nonseparability to demonstrate the strength of space-time coupling, for instance, Phys. Rev. Research 3, 013236 (2021) and OL 44, 2645 (2019). The authors only measured the spatial pattern and spectral chirp, the light field I feel still be space-time separable, not coupled.

The authors didn't retrieve the spatiotemporal structure of light. There is only one very small insert shown in Fig. 6, and I don't know how they reconstruct the spatiotemporal structure based on the existing measured results. The supplement didn't show any more. And I didn't see any retrieved phase distribution.

What is the polarization of the light field? Did the authors study the polarization vector distribution of the light field?

The pulse seems not very ultrafast, with duration about ~ 1 ps. Could the authors comment on the availability to extend their method into fs level or few cycle level?

The references were arranged in a mess, there are repeated items, e.g. ref 12 and ref 36 are same, and many highly related recent advances were not included, e.g. the roadmap J. Opt. 25 093001 (2023), toroidal pulses NC 12, 5891 (2021), NP 16, 523–528 (2022), and spiral pulses NP 17, 822–828 (2023).

The figures were poorly designed, many pieces of panels were shown in many fragmented and incoherent figures. The reader cannot get the logic and story from the arranged figures. I don't think the figure quality meet the NPG criteria.

The inserted patterns in setup figure and Fig S1 are without colorbar.

Minor point: Usually, we use "space-time coupled or coupling" to describe the nonseparable relationship of light structure between space and time domains, rather than "spatiotemporally coupled".

Reviewer #3 (Remarks to the Author):

The manuscript entitled "Direct spatiotemporal manipulation mechanism to generate spatiotemporally coupled ultrafast light fields" by Qinggang Lin et al details their work to demonstrate a direct time-space manipulation method, by introducing a space-dependent time delay and geometrical transformation. The paper is nicely written and constructed in a logical manner, I appreciate that the authors have spent time and effort in conducting adequate mathematical proofs, experiments as well as corresponding analyses. In my opinion, the idea is cool and the work is very interesting, hence I would suggest it can be considered for publication in Nature Communications. I have a few comments for the authors to consider:

1, I would suggest the authors put the Methods part (to be fair it is only 1 Method) either in the Main article (which should be better around the section 'Experimental realization with an exemplary result') or in SI. As a general reader, I found it is really helpful to first get the overall setup to understand the mechanism of 'how to do space-dependent time delay' as well as 'geometrical transformation'.

1.1) In Fig. 6 of the Methods, I would say pls expand the two 'phase figures', they are really small and cannot recognize the phase patterns;

1.2) The colorbar of the beam profiles inside the figure should be added;

1.3) What is the polarisation state of the laser beam output? I think it needs to be clarified;

1.4) I hope the authors could comment more on the beam's polarisation state after the Brewster prism – would it become non-uniform? Would that feature certain effects that are due to Fresnel's effect? If not, what assumptions that we need to make? Please add more discussion of the polarization issue on this beam generation side;

1.5) linked with question 1.4, would there be an issue of the intensity non-uniformity after the P?

1.6) I am just curious about the tilting angle of the Brewster prism – is there any practical best angle for generating the expected beam? What will be its effects on the final experiments? For instance, intensity, phase, and polarisation changes and corresponding side effects.

2, A recent, important review paper is missing in the literature review:
<https://www.nature.com/articles/s41377-022-00897-3>

3, The sizes of Fig.1 to Fig.6 are all too small – pls make them bigger for better visualization.

4, In Fig. 2, why in certain sub-figures, such as figs f, k, l – on the bottom right corner the intensity just strongly lost (interference pattern), is that because the original intensity has already non-uniformed distributed? I can understand this, but I doubt whether it would be lost in such a strong way -- pls give more explanations.

5, For structured light, usually the real-world application is always being criticized, I understand the authors have cited Ref 35-39, as well as in their discussion ‘which may be applicable to optical communications, information encryption, and laser-plasma acceleration’ However, I do hope the authors can give at least one detailed – a unique application that single-coil LS can do/strongly benefit – I hope this can be added in discussion or some places in the paper – which would be very useful in enhancing the paper quality.

6, In SI, the Fig SI needs colorbar, pls add them.

7, Similar figure size issues should be fixed in Fig S4-S6, they are really too small.

8, Pls check/polish all the language throughout the paper.

10th-Dec-2023

Manuscript entitled “Direct space-time manipulation mechanism for spatiotemporal coupling of ultrafast light field”.

Reviewer 1:

This work may have a higher impact in the structured light community and in their broad applications if the authors demonstrate, or make it evident, that the same or similar technique can be applied to sculpture spatiotemporal coupled fields other than the single-coil spring of light to which the whole manuscript is focused, and is not limited to light springs. The authors mention "to construct various complex STC light fields" without further development. For example, why only single-coil and not more coils? What about toroidal pulses or STOVs? Which are the limitations of the method? Depending on this, the claim of the title should be lowered.

Answer: Thanks for your comments. In aspect of generating light springs, generation of single-coil light spring is more complex due to the fact that a single-coil light spring has a continuous topological charge values instead of discrete topological charges carried by multi-coil light spring, which is part of the reason why single-coil light spring is chosen as an experimental demonstration in this study. On the other hand, in order to verify the capability of the system on generating other light fields, we have added simulation results of generating multi-coil light springs, rotating light field with the 8-shape trajectory as well as an annular space-time light field with the compound polarization to Part IV of the supplementary materials. It can be seen that the proposed approach is capable to generate various STC light fields.

- It is a bit heavy to repeat everywhere STC.

Answer: Thanks for your comments. We have revised the entire manuscript accordingly.

- The English language needs to be slightly improved. E.g. "one decade due to their many potential [2, 3], ultrashort pulse velocity control..." does not make sense to me.

Answer: Thanks for your comments. We have polished the language throughout the manuscript.

- The references to springs of light are not properly cited in the text, are disordered, and at least one is missing. The first proposal is not in [10], but in PRL 111, 083602 (2013) in the context of high-harmonic generation, which is not cited. Then, quite speculative theoretical considerations are proposed in [10]. Ref. [3] describes the four-dimensional structure of springs of light including their propagation properties and, in my opinion, cannot be cited only at the very beginning of the text mixed with other references on other topics. Indeed, the authors observe the propagation properties of light springs as predicted in [3]. Please remove "initially proposed in 2015" in the introduction and in the discussion section (which, in passing, is quite repetitive).

Answer: Thanks for your comments. We have revised the introduction part accordingly. We have revised the issue of confusing references and added "PRL 111, 083602 (2013)" as a new reference to the latest version (i.e., the reference [27]), meanwhile added Ref. [3] in the description paragraph of Fig. 6, i.e. “It conforms to the theoretical expectations of the propagation characteristics for light spring in the reference [3]”. Additionally, we have deleted the sentence "initially proposed in 2015"

in the paper.

- All contour plots in the figures lack titles and units. I suppose the titles are x and y, but it must be written. I cannot get an idea of the size of the springs of light without units.

Answer: Thanks for your comments. We have incorporated corresponding size labels for each image in the latest version.

- Other point is that the rotation period is 2150 fs, each frame in Fig. 1 takes 266.7 fs and there are 10 frames, which yields 2667 fs, not 2150 fs. If I'm not wrong, please clarify.

Answer: Thanks for your comments. In Fig.1, we recorded the single-coil spring by pump-probe method with 10 frames. Time interval between two frames is 266.7 fs, which is determined by the step size of the time delay which yields 2667fs in total as mentioned by reviewer. However, rotation period 2.15 ps is obtain by observing the orientation of the spots with respect to the time and estimate time costed for a full 360° rotation by fitting method as shown in Fig. 2 (a). These two values are not in conflicts.

- In short, in addition to the above points, the main point is that the authors should expand on the applicability of their method to synthesize other STC fields.

Answer: Thanks for your comments. The question has been addressed in detail in the first question raised by the reviewer.

Reviewer 2:

- This paper was written toward a very hot topic, the spatiotemporal light field manipulation. The authors present a beautiful spiral light pulse generation and control with very high tunable topological charges, which looks very nice. While, the authors claim they can do direct spatiotemporal manipulation of space-time coupling of ultrashort pulses, I am afraid the authors didn't really realize this. Firstly, the authors didn't measure the space-time coupling of the light fields. To claim the space-time coupled light fields, we need to quantitatively measure the degree of space-time nonseparability to demonstrate the strength of space-time coupling, for instance, Phys. Rev. Research 3, 013236 (2021) and OL 44, 2645 (2019). The authors only measured the spatial pattern and spectral chirp, the light field I feel still be space-time separable, not coupled.

Answer: Thanks for your comments. The paper, Phys. Rev. Research 3, 013236(2021), reported a quantum-mechanics-inspired methodology for quantitatively characterize space-time nonseparability of structured pulses, while the paper, OL 44, 2645(2019), demonstrated that a standard measurement of entanglement, the Schmidt number, determines the propagation distance under which the wave packets retain their shape, and found that reduction in this degree of classical entanglement manifests itself in an increased spread in the measured spatiotemporal spectral correlations. Without doubt, both papers have presented experimental verification approaches for the spatiotemporal correlations. Here, we illustrate the spatiotemporal coupling of output light field by both theoretical and measured data.

Theoretically, the generated single-coil light spring can be expressed as

$$E_{LS}(u, v, t) = \frac{2\pi a A}{b} \exp\left(\frac{r_x^2}{4a^2}\right) \cdot f(u, v) \cdot g(u, v, t) \cdot \exp(i\omega_0 t). \quad (1)$$

with

$$f(u, v) = \exp\left[-\frac{a^2}{r_x^2} \left(\ln \frac{\sqrt{u^2 + v^2}}{b} + \frac{r_x^2}{2a^2}\right)^2\right] \cdot \exp\left[-\frac{a^2}{r_y^2} \arctan^2\left(\frac{v}{u}\right)\right], \quad (2)$$

$$g(u, v, t) = \exp\left[-\alpha \left(t - \eta a \cdot \arctan\left(\frac{v}{u}\right)\right)^2\right], \quad (3)$$

According to Eq. (1-3), the determination of whether the single-coil light spring is spatiotemporal coupling light field mostly relies on $g(u, v, t)$ which can be expressed as

$$g(u, v, t) = \exp(-\alpha t^2) \cdot \exp\left(-\alpha \eta^2 a^2 \cdot \arctan^2\left(\frac{v}{u}\right)\right) \cdot \exp\left[2\alpha \eta a \left(t \cdot \arctan\left(\frac{v}{u}\right)\right)\right], \quad (4)$$

Eq. (4) shows the coordinates of time t and space u and v in $g(u, v, t)$ exhibit power-exponential relations, it is thus impossible to separate t from space u and v in $g(u, v, t)$. Additionally, $g(u, v, t)$ determines that the single-coil light spring has a helical space-time light intensity distribution and the linearly varying topological charge in the frequency domain.

In the experimental measurements, we verify that the generated output light field (as shown in Fig. 1 and Fig. 2) exhibits the structure characteristics of $g(u, v, t)$ with a rotating light spot around the propagation axis at a constant speed, providing direct evidence of its nature as a single-coil light spring with space-time coupling characteristics.

• The authors didn't retrieve the spatiotemporal structure of light. There is only one very small insert shown in Fig. 6, and I don't know how they reconstruct the spatiotemporal structure based on the existing measured results. The supplement didn't show any more. And I didn't see any retrieved phase distribution.

Answer: Thanks for your comment. In the latest version, we have added the measured and simulated results (Fig. 1(c)) of the spatiotemporal intensity for the output field, and the simulated interference results (Fig. 3(b1-b12)) of the output field. Here, the red image of Fig. 1(c) is not as continuous as the cyan image of Fig. 1(c). Due to the limitation of the probe pulse width (172fs), to ensure non-overlapping in time information between two adjacent images, the scanning interval in the experiment was set to 266.7fs. This allowed us only to scan with discrete spatiotemporal sampling. However, the trend of change in both the red image and the cyan image is consistent with our theoretical expectations. Additionally, by comparing Fig. 1(a1-a11) and Fig. 1(b1-b11), Fig. 3(a1-a12) and Fig. 3(b1-b12), it can be seen that the single-coil light spring produced by the experiment has the same time-domain intensity distribution and frequency-domain topological charges distribution as the simulation results. The above results demonstrate that the output field generated represents a single-coil light spring, which is spatiotemporal coupling.

• What is the polarization of the light field? Did the authors study the polarization vector distribution of the light field?

Answer: Thanks for your comment. The light field is linearly polarized throughout the entire

process. In this manuscript, we do not take the vector field into consideration.

- The pulse seems not very ultrafast, with duration about ~ 1 ps. Could the authors comment on the availability to extend their method into fs level or few cycle level?

Answer: Thanks for your comment. According to the expression of the single-coil light spring, the rotation period and topological charge bandwidth are $2\pi\eta a$ and $2\pi\eta a \cdot \Delta\nu$ respectively (as shown in Eq. (6b) and Eq. (7) in the main text). Here, a represents the size coefficient of the geometric transform device, η is the tilt coefficient of the pulse front, and $\Delta\nu$ is the bandwidth of the seed laser pulse. To get a femtosecond-level rotation period, we need a small value of ηa , which can be theoretically realized by reducing the values of η or/and a . The reduction of η can be achieved by abating the angle dispersion introduced by the prism, and the reduction of a can be accomplished by adjusting the structure of GTE and PCE device. Since we are using a laser pulse width of 172fs in this experiment, if the rotation period ($2\pi\eta a$) of the output optical field is less than 172fs, its spiral characteristics will become less apparent and its corresponding topological charge bandwidth ($2\pi\eta a \cdot \Delta\nu$) can be less than 1. At this point, instead of being a single-coil light spring, the output pulse characteristics will resemble vortex light more closely. In order to increase the topological charge bandwidth, we can choose the seed laser with a spectral bandwidth which can support few-cycle pulse duration. Based on the above analysis, this technique can realize the generation of single-coil light springs with fs level or few cycle level by using seed pulses at the fs level.

- The references were arranged in a mess, there are repeated items, e.g. ref 12 and ref 36 are same, and many highly related recent advances were not included, e.g. the roadmap J. Opt. 25 093001 (2023), toroidal pulses NC 12, 5891 (2021), NP 16, 523-528 (2022), and spiral pulses NP 17, 822-828 (2023).

Answer: Thanks for your comment. We have revised the references as well as related part in the manuscript carefully.

- The figures were poorly designed; many pieces of panels were shown in many fragmented and incoherent figures. The reader cannot get the logic and story from the arranged figures. I don't think the figure quality meet the NPG criteria.

Answer: Thanks for your comment. We have revised all figures for better presentation in the new manuscript.

- The inserted patterns in setup figure and Fig S1 are without colorbar.

Answer: Thanks for your comment. We have revised all figures for better presentation in the revised manuscript.

- Minor point: Usually, we use "space-time coupled or coupling" to describe the nonseparable relationship of light structure between space and time domains, rather than "spatiotemporally coupled".

Answer: Thanks for your comment. We think "spatiotemporal coupling" is more appropriate and replaces "spatiotemporally coupled" in the article.

Reviewer 3:

I would suggest the authors put the Methods part (to be fair it is only 1 Method) either in the Main article (which should be better around the section ‘Experimental realization with an exemplary result’) or in SI. As a general reader, I found it is really helpful to first get the overall setup to understand the mechanism of ‘how to do space-dependent time delay’ as well as ‘geometrical transformation’.

Answer: Thanks for your comment. The placement of the Methods part at the end of the article is a requirement of the Nature Communications format and cannot be changed. And to better explain our method, we have added a generating setup description of the single-coil light spring in the supplementary to give a better explanation of the principles of “how to do space-dependent time delay” as well as “geometrical transformation”.

In Fig. 6 of the Methods, I would say pls expand the two ‘phase figures’, they are really small and cannot recognize the phase patterns;

Answer: Thanks for your comment. We have enlarged the sizes of the two phase diagrams in the setup figure and added a corresponding color bar.

The colorbar of the beam profiles inside the figure should be added;

Answer: Thanks for your comment. In the newly revised draft, we have included a corresponding color bar for the beam profiles.

What is the polarization state of the laser beam output? I think it needs to be clarified;

Answer: Thanks for your comment. The polarization direction of the laser output pulse is linear polarization. In the revised manuscript, we have added some explanations. For example, “In our experiments, the light source is a fiber laser which outputs 30.58 MHz/172 fs/1030 nm pulses with linear polarization” and “It can be seen that the spatiotemporal manipulation is continuous, which, consequentially, allows to generate a high-quality single-coil LS with linear polarization” in the second paragraph of the “Experimental realization with an exemplary result” section, and “A fiber laser outputs 30.58 MHz/172 fs/1030 nm pulses with linear polarization” and “The resulting pulse-front tilted pulse has a continuous spatiotemporal variation, which enables generation of high-quality STC light fields with linear polarization” in “Methods” section. In this work, we did not take the vectorial properties of the light field into consideration, polarization is changed during the process only to fulfill the requirement of modulating elements.

I hope the authors could comment more on the beam’s polarisation state after the Brewster prism-would it become non-uniform? Would that feature certain effects that are due to Fresnel’s effect? If not, what assumptions that we need to make? Please add more discussion of the polarization issue on this beam generation side;

Answer: Thanks for your comment. We would like to firstly make it clear that polarization has not been used in the STC light field generation process in this manuscript, in other words, we have not taken vectorial properties of the light field into consideration in this manuscript. The polarization of the beam through the prism would not become non-uniform. In our experimental setup, a Brewster prism is used to generate the tilted pulse front and purify the polarization simultaneously. Due to the fact that liquid crystal devices used for light field modulation here are sensitive to the incident beam polarization, we use an additional polarizer after the prism to further purify the polarization of the beam to get experimental results with higher quality. The Fresnel effect caused by the prism

mainly arises from the difference in refractive index among different wavelength components in the incident pulse. However, this effect can be ignored because there is only a small difference between different spectral components as shown in following calculations.

In experiment, the prism material is H-ZF13, and the top angle is 60° . The spectral values of the incident pulse at 0.1 times the peak position of the laser spectrum are $\lambda_1=1023\text{nm}$ and $\lambda_2=1039\text{nm}$ respectively, which represent the effective boundaries of the laser spectrum. Its refractive index at λ_1 and λ_2 is $n_1=1.752$ and $n_2=1.755$ respectively, and the incidence angle of pulses at the prism incident plane θ_1 is 61.34° which refers to about 2% error (as shown in Fig. 1). When the incident pulse is P-polarized, according to Fresnel formula, the transmittance of the two wavelength components at the incident plane and the exit plane are:

$$T_{p1,in} = \frac{n_1 \cos \theta_{21}}{n_0 \cos \theta_1} \cdot \frac{4 \sin^2 \theta_{21} \cdot \cos^2 \theta_1}{\sin^2 (\theta_1 + \theta_{21}) \cos^2 (\theta_1 - \theta_{21})}, \quad (1a)$$

$$T_{p1,out} = \frac{n_0 \cos \theta_{41}}{n_1 \cos \theta_{31}} \cdot \frac{4 \sin^2 \theta_{41} \cdot \cos^2 \theta_{31}}{\sin^2 (\theta_{31} + \theta_{41}) \cos^2 (\theta_{31} - \theta_{41})}, \quad (1b)$$

$$T_{p2,in} = \frac{n_2 \cos \theta_{22}}{n_0 \cos \theta_1} \cdot \frac{4 \sin^2 \theta_{22} \cdot \cos^2 \theta_1}{\sin^2 (\theta_1 + \theta_{22}) \cos^2 (\theta_1 - \theta_{22})}, \quad (1c)$$

$$T_{p2,out} = \frac{n_0 \cos \theta_{42}}{n_2 \cos \theta_{32}} \cdot \frac{4 \sin^2 \theta_{42} \cdot \cos^2 \theta_{32}}{\sin^2 (\theta_{32} + \theta_{42}) \cos^2 (\theta_{32} - \theta_{42})}. \quad (1d)$$

Here, $T_{p1,in}$ and $T_{p2,in}$ are the transmittance of λ_1 and λ_2 at incident interface, while $T_{p1,out}$ and $T_{p2,out}$ are the transmittance of λ_1 and λ_2 at exit interface, respectively. n_0 is the refractive index of air; $\theta_{21}=\arcsin(\sin(\theta_1)/n_1)$ and $\theta_{22}=\arcsin(\sin(\theta_1)/n_2)$ are the refracting angles of λ_1 and λ_2 in incident surface, $\theta_{31}=\pi/3-\theta_{21}$ and $\theta_{32}=\pi/3-\theta_{22}$ are the incident angle of λ_1 and λ_2 at exit interface, respectively; Accordingly, the refracting angles of λ_1 and λ_2 at exit interface can be figured out by $\theta_{41}=\arcsin(n_1 \cdot \sin(\theta_{31}))$ and $\theta_{42}=\arcsin(n_2 \cdot \sin(\theta_{32}))$, respectively. From the Eq. (S14a-S14d), the transmittance ratio of λ_1 and λ_2 after passing through the prism is

$$\eta_p = \frac{T_{p2,in} \cdot T_{p2,out}}{T_{p1,in} \cdot T_{p1,out}}. \quad (5)$$

Suppose that $\theta_1=61.34^\circ$, $n_1=1.752$ and $n_2=1.755$, the transmittance ratio η_p for P-polarized light is $\sim 99.99\%$. Similarly, the transmittance ratio η_s of λ_1 and λ_2 of S-light is 99.24% . Based on the above analysis, the differences between 1023nm and 1039 nm can be ignored.

Fig. 1 Refraction relationship of the pulses with different wavelengths in Brewster prism. θ_1 is the incident angle at incident interface, θ_{21} and θ_{22} is the refracting angle of λ_1 and λ_2 at incident interface, respectively; θ_{31} and θ_{32} is the incident angle of λ_1 and λ_2 at exit interface, respectively; θ_{41} and θ_{42} is the refracting angle of λ_1 and λ_2 at exit interface, respectively.

linked with question 1.4, would there be an issue of the intensity non-uniformity after the P?

Answer: Thanks for your comment. In the previous answer, we have calculated influence of polarization on intensity uniformity under our experimental conditions, the transmittances of the prism at different wavelengths are almost invariant, so it will not cause the intensity non-uniformity after the P.

I am just curious about the tilting angle of the Brewster prism-is there any practical best angle for generating the expected beam? What will be its effects on the final experiments? For instance, intensity, phase, and polarisation changes and corresponding side effects.

Answer: Thanks for your comment. We select the Brewster prism to get tilted pulse front just because, apart from inducing angular dispersion, it can purify the polarization of its incident pulse, meanwhile has minimal induced loss. Indeed, a prism with different apex angle will induce different angular dispersion η , which allows to get the light spring with different rotation period and the spectral topological bandwidth. Accordingly, apex angle of the prism can be used to change the rotation period and the topological bandwidth of the light spring. However, when the prism has its apex angle away from the Brewster, its induced loss will increase, and the purity of laser pulse from the prism decrease, simultaneously. Usually, we can use a Brewster prism made of different material to get different angular dispersion according to the application requirements.

A recent, important review paper is missing in the literature review: <https://www.nature.com/articles/s41377-022-00897-3>.

Answer: Thanks for your comment. In the latest revised manuscript, we have added this literature.

The sizes of Fig.1 to Fig.6 are all too small-pls make them bigger for better visualization.

Answer: Thank you very much. In the revised paper, we have increased the sizes of each image for better visualization.

In Fig. 2, why in certain sub-figures, such as figs f, k, I - on the bottom right corner the intensity just strongly lost (interference pattern), is that because the original intensity has already non-uniformed distributed? I can understand this, but I doubt whether it would be lost in such a strong way -- pls give more explanations.

Answer: Thanks for your valuable reminder. In our experiments, the seed laser intensity distribution seems OK. In this paper, we find that the intensity loss in the interference patterns only occurs in the measurements at different narrowband spectra. Consequentially, we guess that the intensity losses in the patterns may be associated with the spectrum-space coupling. As is known, the single-coil light spring is generated by folding long elliptical Gaussian pulses, which makes the light spring spatial intensities to be dependent of wavelength or frequency, which makes the case more complicate. Firstly, here, any spectral uniformity can be coupled into spatial intensity distribution. Secondly, when inputting the single-coil light spring into the 4-*f* pulse shaper as show in Fig. S4 (the setup to detect the spectral TC value of the generated single-coil LS), spectrum-space coupling also results in the “chaotic” intensity distribution in the Fourier plane of the 4-*f* imaging system: now, different spectral components with different incident angles to the first grating, so, in the Fourier plane, all spectral components of the pulse don't separate spatially exactly any more.

Consequently, the generated interference patterns coupled with the spatial intensity distribution, will not be the exact spectral interference. Additionally, the dispersions and the fabrications of the two phase-plates made of liquid crystal polymers can also result in the intensity loss of interference modes for some frequency components.

For structured light, usually the real-world application is always being criticized, I understand the authors have cited Ref 35-39, as well as in their discussion' which may be applicable to optical communications, information encryption, and laser-plasma acceleration' However, I do hope the authors can give at least one detailed - a unique application that single-coil LS can do/strongly benefit - I hope this can be added in discussion or some places in the paper - which would be very useful in enhancing the paper quality.

Answer: Thanks for your comment. We added the unique application of the single-coil light spring in the last paragraph of the article introduction section, which is "In particular, the angular delay of the single-coil light spring is expected to serve as the light source for annular on-chip dielectric laser-based accelerators". Under the influence of the driving field, particles released from the ring accelerator will accelerate in a circular motion. However, as the speed of the particles increases, they gradually separate from the driving field, leading to a halt in acceleration. By utilizing a single-coil light spring with linear angular delay as the driving field, it can keep pace with the propagation of the accelerating particles and continuously match the driving field with the accelerating particles [Commun. Phys. 5, 175 (2022)], thereby providing higher acceleration.

In SI, the Fig SI needs colorbar, pls add them.

Answer: Thanks for your comment. We have added corresponding color bars in Fig. S1 of the supplementary materials.

Similar figure size issues should be fixed in Fig S4-S6, they are really too small.

Answer: Thanks for your comment. We have rearranged the image positions in Fig. S4-S6 and increased the size of each image.

Pls check/polish all the language throughout the paper.

Answer: Thanks for your comment. In the new revised version, we have polished the language over the entire manuscript.

Change List

Title section

1. The title "Direct spatiotemporal manipulation mechanism to generate spatiotemporally coupled ultrafast light fields" **becomes** "Direct space-time manipulation mechanism for spatiotemporal coupling of ultrafast light fields".

Abstract section

1. In line 15, "Manipulation of spatiotemporally coupled (STC) light fields..." **becomes** "Manipulating spatiotemporal coupling (STC) of light field...".
2. In line 16, "The manipulation" becomes "At present, such manipulation", "spectrum-space" becomes "space-spectrum".

3. In line 17, “time-space” becomes “space-time”, “for” becomes “, by taking advantages of”, “natures” becomes “nature”, delete “the” after “... natures of”.
4. In line 18, “domains because” becomes “. This is due to the fact that”, “fields” becomes “field”.
5. In line 19, “Spectrum-space” becomes “Although space-spectrum”, “using” becomes “with”.
6. In line 20, “has proved” becomes “has been proved to be”, “but” becomes “it”.
7. In line 21, “, plus information crosstalk from the spectral modulation in spectrum-space plane” **becomes** “. Moreover, information crosstalk during spectral modulation also limits its performance”.
8. In line 22, add “in this manuscript,” before “an innovative mechanism...”, “for” **becomes** “as a”, “time-space” becomes “space-time”.
9. In line 23, delete “an”, “which involves” becomes “by”.
10. In line 24, “by” becomes “via”.
11. In line 25, “As an” becomes “In the”, delete “based on this mechanism”.
12. In line 26, “light spring with STC, as an example,” becomes “STC light spring”, add “based on the mechanism” after “...experimentally generated”.
13. In line 28, add “a distance from near field to far field,” before “with a stable spatiotemporal...”
14. In line 29, **delete** “a basically”, “...from the near field to the far field, which is critically important for its further manipulation and applications. This work provides...” **becomes** “.This is of critical importance for further light field manipulation providing...”.
15. In line 31, “spatiotemporal beams” becomes “STC light fields”.
16. In line 32, “which have” becomes “with potential”.

Introduction section

1. In line 35, “spatiotemporally coupled” becomes “spatiotemporal coupling”.
2. In line 36, “gaining” becomes “attracting”, “...for more than one decade due to their many potential...” **becomes** “...in fundamental studies of space-time light manipulation for more than one decade...”
3. In line 37, “many potential [2, 3]” becomes “potential application [2-8]”, “[4]” becomes “[9]”, “[5]” becomes “[10]”.
4. In line 38, “many STC light fields” becomes “such light fields”, “...other applications. Consequently, many STC...” becomes “...others. Furthermore such...”, add “with potential applications” after “...have been demonstrated”.
5. In line 39 and 40, delete “STC”, “[6]” becomes “[11]”, “[7]” becomes “[12]”, “[8, 9]” becomes “[13, 14]”.
6. In line 41, add “as” before “ultrafast light fields...”, “so quickly” becomes “so rapidly”.
7. In line 42, “...keep pace with them. Consequently, the manipulation of STC light fields...” **becomes** “cope with, manipulation STC of light field...”
8. In line 43, “...spectrum-space domain, rather than in time-space domain, by profiting from...” **becomes** “...the space-spectrum domain indirectly, where the direct space-time manipulation can be avoided due to...”
9. In line 44, “natures of the time” becomes “nature of time”, delete “domains of ultrafast light fields”.
10. In line 45, “A 4-f pulse shaper comprising a pair of gratings and a 4-f imaging system [22, 23] has been widely used to realize spectrum-space manipulations for effective generation of STC

light fields” becomes “In this case, a 4-f pulse shaper comprised by a pair of gratings and a 4-f imaging system [15, 16] has widely been used to realize STC of light fields”.

11. In line 47, “the pulse shaper” becomes “the 4-f imaging system”.
12. In line 48, “[24-26]” becomes “[17-19]”.
13. In line 49, “[6, 8, 27-30]” becomes “[11, 13, 20-23]”. “[31, 32]” becomes “[24, 25]”, “to be introduced to perform” becomes “for”.
14. In line 50, “methodology” becomes “technique”.
15. In line 51, “STC light fields” becomes “STCs of light fields”.
16. In line 52, “[6]” becomes “[11]”.
17. In line 53, “[33]” becomes “[26]”, “[9, 23-26, 30];” becomes “[14, 16-19, 23].”, delete “STC”.
18. In line 54, “unfortunately, however, it also has some intrinsic disadvantages” **becomes** “Although the 4-f pulse shaper has achieved great success, it also encountered a couple of intrinsic problems”.
19. In line 56, delete “issue” add “even more” before “critical when large...”
20. In line 57, delete “STC”.
21. In line 58, add “[27, 28]” after “...single-coil light spring (LS)”.
22. In line 59, “manipulations” becomes “manipulation”.
23. In line 60, “...which can degrade the manipulation quality [34]. Additionally, use of gratings may also be problematic because of their...” **becomes** “...degrading the light field manipulation quality [29] significantly. In addition, the gratings could also be problematic due to...”.
24. In line 62, delete “the”.
25. In line 63, “In this paper,” **becomes** “This paper proposes”, **delete** “that introduces a space-dependent time delay to an ultrafast light field, thereby applies a spatial transformation to construct various complex STC light fields”.
26. In line 65, “This mechanism is implemented by tilting the pulse-front of an ultrafast light field to produce a space-dependent time delay and performing a geometrical transformation for the spatial manipulation” **becomes** “This mechanism first introduces a tilting to the pulse-front of an ultrafast light field to produce a space-dependent time delay and then perform a geometrical transformation to spatially shape the pulse front tilted laser”.
27. In line 68, “...and can avoid the utilization of optical grating which leads to problems mentioned above” **becomes** “...so that it is able to avoid utilizing optical grating and its resultant problems as mentioned above”.
28. In line 70, add “an” after “...by inducing”, add “unique property of” after “...dispersion has a”.
29. In line 71, “and ensures” becomes “, resulting in”, “for” becomes “of”.
30. In line 72, add “of” before “light field”.
31. In line 73, “performed” becomes “implemented”.
32. In line 74, delete “generated”, “which has” becomes “leveraging”.
33. In line 75, “[35-37]” becomes “[30-32]”.
34. In line 76, “[38, 39]” **becomes** “[33, 34]”, “To the best of our knowledge, a single-coil LS with a wide TC bandwidth has not previously been realized experimentally since it was initially proposed in 2015” **becomes** “In particular, the unique angular delay of the single-coil light spring is expected to serve as the light source for annular on-chip dielectric laser-based

accelerators. To the best of our knowledge, a single-coil LS with a wide TC bandwidth has not previously been realized experimentally”.

35. In line 79, add “of” before “fields...”

Results section

1. In line 83, **delete** “our”, “...manipulate spatially the light field whose spatial and temporal information are coupled with each other. To realize this manipulation, the first task is to establish a time-space correlation and then perform some spatial transformations of the target STC light fields. For simplicity, in this work, the correlation...” **becomes** “...firstly create space-time correlation for a light field and then perform corresponding spatial transformations for the targeted STC light field. For simplicity, in this work, the first step...”
2. In line 88, “aims to” becomes “would”.
3. In line 90, “this field can be described by” becomes “the field can then be described as”.
4. In line 92, “at a central frequency ω_0 , and $\alpha = C/\tau_p^2$, is determined by a” becomes “, $\alpha = C/\tau_p^2$ is determined by”.
5. In line 94, delete “a”, delete “of”, “y-dependent time-delay” becomes “time-delay along y-axis”.
6. In line 96, delete “generally”.
7. In line 100, “dispersion” becomes “dispersive”.
8. In line 102, “[40]” becomes “[35]”.
9. In line 104, delete “then”, “an” become “the”.
10. In line 108, delete “Equation (3) implies that a large value of the y-direction beam size r_y can increase the maximum of the available space-dependent time delay”.
11. In line 111, “set” becomes “positioned”.
12. In line 113, “[41]” becomes “[36]”.
13. In line 114, “designed” becomes “expressed”.
14. In line 116, “time-space” becomes “space-time”.
15. In line 118, “...spectrum-space domain (details are provided in Part I of the Supplementary Material)” becomes “...space-spectrum domain (see details are provided in Part I of the Supplementary Material)”.
16. In line 119, “temporal” becomes “time”.
17. In line 126, “Equation (4)” becomes “Eq. (4)”.
18. In line 127, “time-space” becomes “space-time”, “Equation (6a)” becomes “Eq. (6a)”.
19. In line 128, “...Eq. (6b) presents a linearly azimuth angle-dependent time delay” becomes “Equation (6b) presents an azimuth angle-dependent linearly time delay with a rotation period of $2\pi\eta\alpha$ ”.
20. In line 129, **add** “The exponential expansion term in Equation (6b) contains a product of time and space information that cannot be separated further. It illustrates that single-coil LS is a spatiotemporal coupling light field” **after** “...time delay.”, “Equation (5)” becomes “Eq. (5)”, “the light field has” becomes “that”.
21. In line 130, “spiral structures” becomes “of the light field has spiral like structures”.
22. In line 131, delete “a”, add “values with a TC bandwidth of $2\pi\eta\alpha\Delta\nu$ ” after “spectrum-dependent TC”.

23. In line 133, “our” becomes “the proposed”.
24. In line 134, “There” becomes “At the moment”, delete “one-dimensionally”.
25. In line 136, **add** “which would lead to the coupling between one-dimensional space and time, and” **before** “then a log-polar-Cartesian”, “one-dimensional space coupling into to” becomes “one-dimensional space and time coupling”.
26. In line 137, add “and time” after “time coupling”.
27. In line 138, “our” becomes “the”.
28. In line 139, add “linearly polarized” after “...1030 nm”.
29. In line 140, “of” becomes “,”, “are confirmed experimentally” becomes “have been experimentally verified separately”.
30. In line 142, “We can see that our spatiotemporal manipulation is continuous, which, consequentially, allows to generate a high-quality single-coil LS” **becomes** “From Fig. S1, it can be seen that the spatiotemporal manipulation is continuous, which, consequentially, is one of key factors for the generation of high-quality single-coil LS”.
31. In line 144, “Figure 1(a1)–(k1) present the intensity distributions of the output pulse from the log-polar-Cartesian transformation system vs. time recorded with an associated time interval of 266.7 fs” **becomes** “Fig. 1(a1–a11) present the spatial intensity distributions of the output light field by pump-probe measurement (refer to Fig S4 in Part II of supplementary for details) with a time interval of 266.7 fs to sample the light field. Here, the time interval can be changed by adjusting the step size of the time delay line”.
32. In line 146, “It is observed that the measured beam spot rotates counterclockwise along the annular trajectory of time” **becomes** “It can be observed that, the experiment results exhibit the same time-varying trajectories as the calculated, specifically, both the beam spots rotate counterclockwise, which is consistent with the temporal variation properties given by Eq. (4)”.
33. In line 147, “The spot brightness also increases over the interval from 0° to 197°, as shown from Fig. 1(a1) to Fig. 1(g1), and then attenuates monotonically up to 360°, as shown from Fig. 1(g1) to Fig. 1(k1). The evolution of the rotation angle vs. time is fitted as indicated by the red line in Fig. 1(m), from which the rotation speed is estimated to be 2.92 rad/ps, where the rotation period Δt is 2.15 ps. The seed pulse duration τ_p is measured to be 172 fs, as shown in Fig. 2(n), and thus $\Delta t \geq \tau_p$. Figure 1(a2)–(k2) and the blue line in Fig. 1(m) represents the corresponding numerical simulation result. We can thus see that the measurements and the theoretical predictions have excellent agreement” **becomes** “Fig. 1(c) shows the measured spatiotemporal intensity evolution (red, constructed from Fig. 1(a1-a11)) and the simulation results (cyan). It can be seen that the measured results correspond well with calculations, which confirms the success generation of the single-coil LS. However, it worth point out that, the red image is discrete, which can be attributed to the scanning time interval of the pump-probe measurement (266.7fs) is chosen to be larger than the pulse width (172fs) to avoid overlapping between neighbored frames. As mentioned previously, one can choose a shorter time interval by adjusting the delay time to sample spatiotemporal structure of the light field for finer details in time domain. However, a short time interval can make the pump-probe measurement time-consuming, moreover it will not improve the sampling quality. The good matching between the red and cyan results in Fig. 1(c) indicate that a time interval of 266.7fs is short enough to sample effectively the single-coil LS with a rotation period of 2.15ps”.

34. In line 157, **the figure illustrates** “Fig. 1. Measured (a1)–(I1) and simulated (a2)–(I2) spatial and temporal intensity distributions of the output from the log-polar-Cartesian transformation system, (m) the rotation angle of the output pulse intensity with time, and (n) the measured autocorrelation track of the femtosecond pulse from the 1030 nm fiber laser” **becomes** “**The spatial intensity distributions of the generated light field. a1–a11** The experimental measured spatial intensity distribution at different time. **b1–b11** The corresponding calculated results. **a12, b12** The measured and calculated spatial intensity distribution after time domain integrations respectively. **c** The measured (red) and calculated (cyan) spatiotemporal structure of the generated light field”.
35. In line 161, “Figure 1(I1) and Fig. 1(I2) show the time integrals of the spatial intensities from the experimental detection and the theoretical simulation, respectively” becomes “Fig. 1(a12) and Fig. 1(b12) show the measured and calculated spatial intensity distribution of the generated light field after time integration, respectively”.
36. At end of line 166, add a paragraph “To further quantitatively explore the properties of this light field, the evolution of the rotation angle vs. time is plotted in Fig. 2(a), red squares are extracted from experiments, while red line is a linear fitting of the experiment results. Blue squares and line are corresponding calculated results. From Fig. 2(a), the rotation speed can be estimated to be 2.92 rad/ps, and the rotation period Δt is thus determined as 2.15 ps, which corresponds to the total time needed for a 360° rotation of the light spot in azimuthal direction. Moreover, it can be seen that the rotation period is larger than the seed pulse duration τ_p which is measured to be 172 fs due to dispersion introduced during modulation, as shown in Fig. 2(b). Additionally, from the autocorrelation profile, it can be observed that the seed pulse contains a picosecond background, which, together with optical diffraction, results in occurrences of the background noise in Fig. 1 (a1–a12). The blue squares and line in Fig. 2(a) represent the corresponding calculation results, which agrees well with the measurements. Based on the above results, it can be concluded that the measurements and the theoretical predictions have excellent agreement, which indicates a success generation of a single-coil LS experimentally. According to Eq. (6b), the spatiotemporal information of a single-coil LS cannot be separated, i.e., the LS is space-time coupled. Experimentally, time-varying spatial intensity distribution of the single-coil LS has been measured as well and shown in Fig. 1(c), it can further confirm STC characteristics of the generated LS”.
37. In line 167, “Figure 2(a)–(l) aim to” becomes “In this section, measurements are made and plotted in Figure 3(a1–a12) to”.
38. In line 168, “...of the log-polar-Cartesian transformation system and the 1030 nm femtosecond probe beam” **becomes** “and the femtosecond 1030 nm probe as shown in Fig. S4”, “Here, twelve narrowband spectra with a bandwidth of ~1.20 nm are selected by moving the slit to vary the central wavelength from 1023 nm to 1039 nm by a step of ~1.46 nm” **becomes** “By moving the slit position, twelve narrowband spectra are selected, which have their central wavelengths from 1023nm to 1039 nm with an interval of ~1.46 nm and a bandwidth of ~1.20 nm as shown in Fig. 3(c)”.
39. In line 171, “The results are extracted as shown in Fig. 3(m) (see also Fig. S3 in the Supplementary Material). The results show that the spectral TC value varies from –6 to –17 as the wavelength varies from 1023 nm to 1039 nm, where the central TC, or the TC at the central wavelength has a value of –11” **becomes** “From Fig. 3(a1–a12), the spectral TC values

have been extracted, which varies from -6 to -17 with the wavelength from 1023 nm to 1039 nm as shown by the red dots in Fig. 3(d). The central TC, or the TC at the central wavelength can be estimated at -11 ".

40. In line 175, "Figure 3(n) shows the TC distribution as a function of wavelength, where the red dots represent the measured values and the red line is the linear fitting of the seed spectrum represented by the blue line. The figure shows that the spectral TC value evolves linearly with the wavelength from around 1030 nm; equivalently, it evolves linearly with a frequency of approximately 1.83×10^{15} Hz when $\Delta\omega \ll \omega_0$. The corresponding TC varies at a speed of $\Delta/\Delta\omega = 0.381$ THz $^{-1}$. The TC bandwidth is 11.5 over a spectral range with an intensity greater than 0.1 times the peak intensity, as indicated by the blue line in Fig. 2(n)" **becomes** "Figure 3(d) also presents the spectrum of the seed laser by the blue curve and the theoretical fitting of the red dots by the red line. One can see that the spectral TC values have a good linear dependence with wavelengths within the effective spectral region of the seed laser. This important property is consistent with the prediction of the spectral TC variation by Eq. 5. The spectral TC changes by a speed of $\Delta/\Delta\omega = 0.381$ THz $^{-1}$, and the TC bandwidth is as broad as 11.5 . The corresponding simulations have presented in Fig. 3(b1-b12) and the cyan dots of Fig. 3(d), and the cyan line is theoretical fitting of the cyan dots. The spectral structure shown by the blue line in Fig. 3 (d) is also utilized as the seed spectrum of the simulation. It can be concluded that the spectral TC varies linearly by a speed $\Delta/\Delta\omega = 0.390$ THz $^{-1}$, and according to the TC bandwidth is 11.3 . Accordingly, the spectral TC variation obtained from the experiment is consistent well with theoretical expectation. Based on the linear dependence of the spectral TC on wavelength, it can be further demonstrated that the output field constitutes a single-coil LS".
41. In line 183, change the image color and picture number, change the image name from Fig. 2 to Fig. 3, increase the new simulation results Fig. 3 (b1-b12) and Fig. 3 (d), increase a size scale.
42. In line 184, **the figure caption** "Fig. 2. Interference patterns (a)–(l) between the output pulse of the log-polar-Cartesian transformation system and the femtosecond probes with the selected central wavelength, (m) narrowband spectra, and (n) spectral TCs extracted from the interference patterns (red line and marks) and the spectrum of the seed pulse (blue line)" **becomes** "**Fig. 3 Interference patterns between the output from the log-polar-Cartesian transformation system and the femtosecond probes at twelve selected narrowband spectra. a1–a12** The experimental and **b1–b12** simulated interference patterns at different narrowband spectra. **c** Measured narrowband spectra selected by the slit at different position. **d** Spectral TCs (red: measured, crayon: calculated) vs. wavelengths with the seed pulse spectrum (blue): the square marks for the results extracted from the interference patterns and the lines for corresponding linear fittings, respectively".
43. In line 188, add "In this section, it is demonstrated that the central topological charge value is tunable" before "Figure 3(a)–(n)...", "Figure 3(a)–(n)" becomes "Figure 4(a)–(n)", delete "the" after "...interferograms of".
44. In line 189, "which shows the generated LS is tunable with its central TC. In Fig. 3(o)The corresponding central TC values are plotted as the hollow red dots and fitted theoretically and shown by the blue line" **becomes** "which shows the central TC value of the generated LS can be tuned by changing the position of the PCE plate. Fig. 4(o) plots corresponding central TC

values as a function of PCE position (hollow red dots), and blue curve is a linear fitting of these values”.

45. In line 192, “...of the PCE by 1 of the TC value per 44 μm , the shift of PCE position, ...” becomes “...of the PCE. More precisely, the TC value changes by 1 with 44 μm displacement of PCE plate, ...”.
46. In line 194, “using” becomes “by”.
47. In line 195, change the image color, change the image name from Fig. 3 to Fig. 4, increase a size scale.
48. In line 196, **the figure caption** “Fig. 3. Central topological charge of the LS with respect to the PCE position. (a)–(n) Measured interference patterns at the different PCE positions, and (o) the corresponding central TCs of the interference patterns by red hollow points with the blue line for its theoretical fitting.” **becomes** “**Fig. 4 Central topological charge of the LS with respect to the PCE position along y-direction. a–n** Measured interference patterns with different PCE positions. **o** Measured central TCs of the interference patterns as function of PCE position by the red hollow points are extracted from the interference patterns (a–n), and the blue line corresponds to the fit curve”.
49. In line 199, change the color of the point and the line from green to magenta, change the image name from Fig. 4 to Fig. 5.
50. In line 200, **the figure caption** “Fig. 4. Spectral TC values of the single-coil LS vs. wavelength at different y-directional PCE positions” becomes “**Fig. 5 Spectral TC values of the single-coil LS vs. wavelength for different central TCs (2, 0, –3, and –11)**”.
51. In line 202, “green” becomes “magenta”.
52. In line 204, “Our experiments show when we tune the central TC, the LS keeps its spectral TC bandwidth invariant almost” becomes “In this section, it is demonstrated when tuning the central TC by changing the position of the PCE plate along y-direction, the LS keeps its spectral TC bandwidth nearly invariant”.
53. In line 205, “Figure 4” become “Fig. 5”.
54. In line 208, “at various central TCs (2, 0, 3, and 11)” becomes “for different central TCs (2, 0, –3, and –11)”.
55. In line 208, “green” becomes “magenta”, delete “are used to”, “the TC distributions at the” becomes “LSs with”.
56. In line 209, “with” becomes “, and”, “change rates $\Delta I/\Delta \omega$ given by” becomes “values varies at speed of ($\Delta I/\Delta \omega$) of”.
57. In line 210, add “in these cases” before “respectively”, delete “,” before “respectively”.
58. In line 212, add “nearly” before “no change”.
59. In line 213, change the image color and picture number, change the image name from Fig. 5 to Fig. 6.
60. In line 214, **the figure caption** “Fig. 5. Recorded intensities of the generated single-coil LS vs. time at the focal plane (a1)–(I1) and 2 cm behind the focal plane (a2)–(I2), and the corresponding integration intensities (m1) and (m2) over time, respectively” becomes “**Fig. 6 The Spatiotemporal intensity distributions of the generated light field after different propagation distance. a1–a11, b1–b11** Recorded spatial intensities of the generated single-coil LS as a function of time at the focal plane and 2 cm behind the focal plane, respectively. **a12, b12** The corresponding spatial intensities distribution after integration over time”.

61. In line 217, “Figure 5 focuses on the propagation stability of the LS by checking the spatial-temporal intensity distributions in both the near-field and the far-field” becomes “At the end, the propagation properties of the LS have also been investigated. Fig. 6 shows the measured results of spatiotemporal intensity distributions of generated LS in both near-field and far-field”.
62. In line 218, “Figure 5(a1)-(l1)” becomes “Figure 6(a1-a11)”.
63. In line 219, “the far-field spatial-temporal” becomes “the spatiotemporal”, “Fig. 5(a2)-(l2)” becomes “Fig. 6(b1-b11)”.
64. In line 221, “two positions are” becomes “two planes are estimated to be”.
65. In line 223, “...shown in Fig. 1” **becomes** “...of the near field”, **add** “It conforms to the theoretical expectations of the propagation characteristics for light spring in the reference [3].” **before** “Figure 5(m1)”, “Figure 5(m1) and 5(m2) show the measured and simulated integration intensities over time. When compared with the near field, the intensity gap at the bottom of the ring is narrower at the plane 2 cm behind the focal plane, and the gap almost disappears at the focal plane, which can be attributed to optical diffraction” **becomes** “Fig. 6(a12) and 6(b12) show the measured and calculated spatiotemporal intensity after integration over time. When comparing these two cases, it can be observed that, an intensity gap appears after at the bottom of the ring at the plane 2 cm behind the focal plane, which can be attributed to optical diffraction”.

Discussion section

1. In line 231, add “aforementioned” before “experimental”, delete “above”, “our” become “the proposed”.
2. In line 232, “...can work well to realize STC LS” becomes “...works well to realize STC LS, which is previously unachievable with other methods”, “Figure 1 presents its spring-like amplitude, while Fig. 2 shows that the LS owns its spectral TC value linearly dependent on wavelength, so it...” **becomes** “Fig. 1 presents its spring-like amplitude, while Fig. 3 shows that spectral TC values of generated LS vary linearly with wavelength, which is one of the key unique properties of single-coil LS. Moreover, the generated LS”.
3. In line 236, “According to Eqs. (4)–(7), we believe that the generated LS is of single-coil LS. From Fig. 3 and 4, we can see the single-coil LS can be tuned continuously with its central TC value by changing the ν -directional position of the PCE with no change of its spectral TC bandwidth, which...” **becomes** “From Fig. 4 and Fig. 5, it can be seen that the central topological charge value of the single-coil LS can be tuned continuously by changing the position of the PCE plate along y -directional while the TC spectral bandwidth is maintained during this manipulation. This behavior...”
4. In line 240, “As we know” becomes “As known”.
5. In line 241, “We have also measured the intensity structures and the rotation periods of the single-coil LS in both the near field and the far fields (see Fig. 5), which demonstrates excellent propagation stability” **becomes** “Interestingly, the measurement results as shown in Fig. 6 have verified that both the intensity structures and rotation periods of the single-coil LS have excellent propagation stability”.
6. In line 244, “sum” becomes “summary”.
7. In line 245, “tuned” becomes “tunable”.
8. In line 246, delete “since proposed theoretically in 2015”.

9. In line 247, “our” becomes “the proposed”.
10. In line 249, “our design manipulates continuously an ultrafast light field, instead of the separate spectral sampling, ...” **becomes** “instead of the separate spectral sampling, the proposed mechanism manipulates continuously an ultrafast light field, ...”
11. In line 251, add “the” before “clear apertures”.
12. In line 252, “...set at the confocal plane, and the information crosstalk caused by optical diffraction of the spectral modulation pixels” **becomes** “..., as well as information crosstalk caused by optical diffraction during phase modulation”.
13. In line 254, add “Additionally,” before “The setup...”
14. In line 255, “to be easy to align” becomes “ease the alignment”.

Methods section

1. In line 264, “Figure 6” becomes “Fig. 7”.
2. In line 266, add “with linear polarization” after “...1030nm pulses”.
3. In line 267, add “which has negligible induced loss for the linearly polarized incident pulses. After the prism, a polarizer is used to purify further the linear polarization. In order to...” after “...Brewster prism (H-ZF13)”.
4. In line 269, “..., respectively, are used to expand the y direction beam size a (full width at half maximum, FWHM) of the pulses to 18 mm from 1.8 mm, so the maximal y -dependent time-delay $2\pi\eta a$ is about 1.51 ps. Here, ...” becomes “...are used to expand the beam size of the pulses from 1.8 mm to 18 mm (full width at half maximum, FWHM) along y -direction. After that, ...”.
5. In line 271, delete “then” after “(QW1)”.
6. In line 272, “...linear polarization into right-circular polarization” becomes “...linear to circular polarization”.
7. In line 273, “are made from” becomes “made of”.
8. In line 274, add “, are” before “located at”.
9. In line 280, “...circular polarization back to the linear polarization” becomes “circular back to the linear but orthogonal to that of the seed pulses”.
10. In line 282, add “with linear polarization” after “STC light fields”.
11. In line 283, “can be designed to phase-modulate” after “can be designed to further modulate”.
12. In line 284, add “laser” before “pulse for the generations...”
13. In line 286, optimize the image, enlarge the size of the illustration, and add three projection of a single-coil light spring.
14. In line 287, **the figure caption** “Direct space-time manipulation setup to generate single-coil LS from an ultrafast light field” **Becomes** “**Fig. 7 Setup based on direct space-time manipulation to generate single-coil LS from an ultrafast light field**”.
15. In line 289, add “The 3D image of the single-coil LS includes its projection images in the x-y, t-y, and x-t planes” at the end.

References section

1. At the end of line 304, add five new references: **reference 4** “Y. Shen, Q. Zhan, L. G. Wright, D. N. Christodoulides, F. W. Wise, A. E. Willner, K. Zou, Z. Zhao, M. A. Porras, A. Chong, C. Wan, K. Y. Bliokh, C. Liao, C. Hernández-García, M. Murnane, M. Yessenov, A. F. Abouraddy, L. J. Wong, M. Go, S. Kumar, C. Guo, S. Fan, N. Pappasimakis, N. I. Zheludev, L. Chen, W. Zhu, A. Agrawal, M. Mounaix, N. K. Fontaine, J. Carpenter, S. W. Jolly, C. Dorrer, B. Alonso,

I. Lopez-Quintas, M. López-Ripa, Í. J. Sola, J. Huang, H. Zhang, Z. Ruan, A. H. Dorrah, F. Capasso and A. Forbes, “Roadmap on spatiotemporal light fields,” *J. Opt.* 25, 093001 (2023)”, **reference 5** “Y. Shen, Y. Hou, N. Papasimakis, and N. I. Zheludev, “Supertoroidal light pulses as electromagnetic skyrmions propagating in free space,” *Nat. Commun.* 12, 5891 (2021)”, **reference 6** “A. Zdagkas, C. McDonnell, J. Deng, Y. Shen, G. Li, T. Ellenbogen, N. Papasimakis, and N. I. Zheludev, “Observation of toroidal pulses of light,” *Nat. Photonics* 16, 523-528 (2022)”, **reference 7** “M. Piccardo, M. de Oliveira, V. R. Policht, M. Russo, B. Ardingi, M. Corti, G. Valentini, J. Vieira, C. Manzoni, G. Cerullo, and A. Ambrosio, “Broadband control of topological–spectral correlations in space–time beams,” *Nat. Photonics* 17, 822-828 (2023)”, **reference 8** “C. He, Y. Shen, and A. Forbes, “Towards higher-dimensional structured light,” *Light-sci. Appl.* 11, 205 (2022)”.

2. In line 305-317, the serial number of reference 4-9 becomes “9-14”.
3. In line 318, reference 11 becomes reference 28.
4. In line 320-350, delete reference 11-21.
5. In line 351-382, the serial number of reference 22-33 becomes “15-26”.
6. At end of line 382, **Add reference 27** “C. Hernández-García, A. Picón, J. S. Román, and L. Plaja, “Attosecond extreme ultraviolet vortices from high-order harmonic generation,” *Phys. Rev. Lett.* 111, 083602 (2013)”.
7. In line 383-404, the serial number of reference 34-41 becomes “29-36”.

REVIEWERS' COMMENTS

Reviewer #1 (Remarks to the Author):

The authors have satisfactorily addressed my comments, particularly that regarding the applicability of the method to synthesize other spatiotemporal coupled fields. They demonstrate this point by means of numerical simulations in Part 4 of the supplemental material. Still, English should be improved further.

Reviewer #2 (Remarks to the Author):

The authors did careful revisions and responded for all my prior comments. I am generally happy with all the responses and happy to recommend current version to be published after cleaning some minor issue. The authors added analysis of space-time nonseparability, that is good, but missed to cite related papers, the Phys. Rev. Research 3, 013236 (2021) and OL 44, 2645(2019); Another suggestion is that the authors can emphasize the potential application of information encoding using space-time coupling, e.g. ACS Photonics 2023, 10, 7, 2149–2164; Fig. 1c should have legends to show what are the meanings of the blue and red profiles; In Fig.7 the colorbars for intensity were missing.

Reviewer #3 (Remarks to the Author):

I think the authors have addressed my comments properly and the paper is ready for publication.

Reviewer 1:

- The authors have satisfactorily addressed my comments, particularly that regarding the applicability of the method to synthesize other spatiotemporal coupled fields. They demonstrate this point by means of numerical simulations in Part 4 of the supplemental material. Still, English should be improved further.

Answer:

Thanks for your comment. In the new revised version, we have polished the language of entire manuscript.

Reviewer 2:

- The authors did careful revisions and responded for all my prior comments. I am generally happy with all the responses and happy to recommend current version to be published after cleaning some minor issue. The authors added analysis of space-time nonseparability, that is good, but missed to cite related papers, the Phys. Rev. Research 3, 013236 (2021) and OL 44, 2645(2019); Another suggestion is that the authors can emphasize the potential application of information encoding using space-time coupling, e.g. ACS Photonics 2023, 10, 7, 2149–2164; Fig. 1c should have legends to show what are the meanings of the blue and red profiles; In Fig.7 the colorbars for intensity were missing.

Answer:

Thanks for your comment. We have included these two papers in the analysis section on space-time nonseparability (i.e., reference [38, 39]), and we have added the potential application of single-coil LS in information coding along with the corresponding references. Additionally, we have provided the missing legends for Fig. 1c and colorbars for Fig. 7.

Reviewer 3:

- I think the authors have addressed my comments properly and the paper is ready for publication.

Answer:

Thanks for your comment.